# Microwave Transmittance Technique Using Microstrip Patch Antennas, as a Non-Invasive Tool to Determine Soil Moisture in Rhizoboxes [note 1]

**DOI:** 10.3390/s20041166

**Published:** 2020-02-20

**Authors:** Paulo Sergio de Paula Herrmann, Viktor Sydoruk, Felipe Nieves Marques Porto

**Affiliations:** 1Embrapa Instrumentation, São Carlos (SP) 13560-970, Brazil; 2Forschungszentrum Jülich/IBG-2: Plant Sciences, 52425 Jülich (NRW), Germany; v.sydoruk@fz-juelich.de; 3Maastricht Science Programme, Faculty of Health, Medicine, and Life Sciences, Maastricht University—Kapoenstraat 2, 6211 KW Maastricht, The Netherlands; porto.felipe94@gmail.com

**Keywords:** microwave technique, transmittance, soil moisture, microstrip patch antennas, rhizobox, roots, plant phenotyping

## Abstract

Investigating the growth behavior of plant root systems as a function of soil water is considered an important information for the study of root physiology. A non-invasive tool based on electromagnetic wave transmittance in the microwave frequency range, operating close to 4.8 GHz, was developed using microstrip patch antennas to determine the volumetric moisture of soil in rhizoboxes. Antennas were placed on both sides of the rhizobox and, using a vector network analyzer, measured the S parameters. The dispersion parameter S_21_ (dB) was also used to show the effect of different soil types and temperature on the measurement. In addition, system sensitivity, reproducibility and repeatability were evaluated. The quantitative results of the soil moisture, measured in rhizoboxes, presented in this paper, demonstrate that the microwave technique using microstrip patch antennas is a reliable, non-invasive and accurate system, and has shown potentially promising applications for measurement of rhizobox-based root phenotyping.

## 1. Introduction

New tools and approaches are considered important for the investigation and evaluation of soil–water–plant interactions in high-throughput plant phenotyping investigations [1]. A central parameter for determining root system response to water availability is that water is usually not homogeneously distributed in the soil and the heterogeneity significantly increases when drought stress occurs. Thus, the development of non-invasive instruments and sensors to measure soil moisture distribution would open up new approaches to investigate plant strategies to deal with low water content or, in particular, heterogeneities in water availability of soils during periods of drought cycles.

Electromagnetic soil water content sensors are now widely accepted for soil water content determination because these sensors allow continuous, fast, stable and nondestructive sensing of the spatial-temporal dynamics of soil water content at the lab and field scale [2].

From the electromagnetic point of view, the soil–plant–water set is considered a mixture of five dielectric compositions consisting of air, soil volume, bound water and free water and roots. The water molecules that are attached to the soil particles interact with incident electromagnetic waves differently than the free water molecules resulting in different dielectric dispersion spectra. The dielectric constants in the complex form of free and bound water are functions of the electromagnetic frequency (f) [3], the physical temperature (T) [4] and the salinity (S) [5].

The dielectric properties of wet soil are characterized by frequency dependence as a function of dielectric constant response, ε_r_. In soil–plant–water systems the ε_r_ values for the soil are typically between 3 and 8. In free water, also known as bulk water, the values are around 79 at 20 °C, and for the roots of a plant they vary between 42 and 56. The volumetric water contents of the root samples vary linearly with the volume. The ε_r_ of the air is equal to 1. Thus, relatively small amounts of free water in the soil will greatly affect its electromagnetic properties (high degree of polarization under an external electrical field) [6,7].

For non-invasive determination of volumetric soil moisture, θ_V_, a microwave system based on transmittance of electromagnetic waves in the frequency range close to 4.8 GHz was developed using microstrip antennas (MSAs). MSA are a highly developed and widely used technology [8] that has numerous advantages, such as low weight, small volume and ease of fabrication using printed-circuit technology, which led to the design of several configurations for various applications. The example of such applications is presented by Toba and Kitagawa in 2011, where they utilized a wireless moisture sensor that uses backscatter characteristics of the microstrip antenna [9]. Another one is the one demonstrated by Sarabandi and Li in 1997, which is a prototype of the microstrip ring resonator that operates at L-band and is used for measurements of soil with different moisture contents [10]. In this article we used two MSAs with a simple design that were placed on both sides of a rhizobox. The evaluation is made using scattering parameters (known as S-parameters) describing dispersion and dielectric properties of a soil–water system. Figure 1 illustrates the basic principle of the electromagnetic wave (EM) interactions with the matter.

Calibration curves for four porous media are presented, i.e., three for different soils: Nullerde (Einheitserde Typ 0, Einheitserde- und Humuswerke Gebr. Patzer GmbH and Co. KG, Sinntal-Altengronau, Germany), a peat–sand–pumice substrate (Dachstaudensubstrat SoMi 513, Hawita GmbH, Vechta, Germany) and Cerrado soil (tropical soil); and one for glass beads (particles size 0.5 mm). The results presented in this work show a potential of using microwave transmittance technique and MSA for development of an affordable and new non-invasive determination of volumetric soil moisture, θ_V_, which can possibly be applied for a better understanding of root growth in the new field of plant phenotyping.

## 2. Materials and Methods

This section presents all the methodologies and materials involved in conducting the investigation. In Section 2.1 there are details of the vector network analyzer (VNA) that was used, as well its configuration and basic principles of the system. Section 2.2 shows the details of draw, the configuration and the experimental evaluation of the microstrip patch antenna (MSA) developed in our investigation. The geometry of the rhizobox, normally used to study the development of root growth, is shown in Section 2.3. The evaluation of the reproducibility and repeatability of the measurements done, where the dielectric properties were established to emulate the soil–water system, is presented in Section 2.4. The system developed was applied to measure the volumetric soil moisture, and in Section 2.5 the calibration curve for four different porous media (PM), as well the physical–chemistry properties of these samples, is presented. The experiments were conducted in the standard laboratory conditions, without an anechoic chamber.

### 2.1. Vector Network Analyzer (VNA)

The antennas are connected to the Vector Network Analyzer (VNA; ZNB 8, Rohde and Schwarz, Germany), which is generally used to generate and measure radio frequency/microwave signals [12]. Some features about the equipment used a frequency range from 9 × 10^−6^ GHz– 8.5 GHz, wide dynamic range of up to 140 dB and high temperature stability 0.01 dB/°C. Using VNA, scattering parameters (S-parameters, i.e., S_21_ in dB) were measured in the frequency range from 4.6 to 5.0 GHz to characterize transmission of microwaves through a porous medium with a resolution of 0.27 dB. The 4.8 GHz was chosen as a compromise between resolution and penetration depth of electromagnetic waves into water. Tsang et al. [13] evaluated the effect of optical depth of the soil as a function of moisture content for frequencies of 0.25 GHz, 1.42 GHz, 5.0 GHz and 10.69 GHz. They observed that at high frequencies, the optical depth is smaller and has stronger dependence on the moisture content.

A block diagram of the setup that was developed to conduct the experiments presented in this work can be seen in Figure 2. It essentially consists of five parts: (1) the Vector Network Analyzer (VNA) to measure scattering parameters; (2) electro-mechanical positioning systems X and Y to control the position of antennas at the both sides of a rhizobox; (3) two microstrip patch antennas designed to transmit and receive the microwave signal; (4) cables and connectors and (5) software to control the positioning system and to collect data.

### 2.2. Design of Microstrip Patch Antennas

The antennas used in this experiment were designed as λ/2-resonant microstrip antennas (MSAs) [8,10] and executed on a circuit board (glass-reinforced epoxy laminate; FR-4) with 1.5 mm thickness and dielectric constant, ε_r_ of 4.4. Although each antenna has a patch of 15.7 mm wide and 15.4 mm long, the total size was 25.0 × 25.0 × 1.5 mm^3^. Such a design allows one to have effective operation at frequencies close to 4.8 GHz. Figure 3 illustrates our microstrip antenna design for microwave range [8,14].

The dependence of the scattering parameters on the frequency for the MSAs was obtained under the standard laboratory atmospheric conditions (Figure 4). The spectrum was measured in the frequency range between 4.650 and 4.950 GHz with a sweep of 1 MHz. For the 24 mm distance between antennas, the obtained resonant frequency f_0_ was around 4.864 GHz and the magnitudes of S_21_, S_11_ and S_22_ at f_0_ were −10.868 dB, −24.486 dB and −15.824 dB, respectively.

Additionally to that, the antennas were evaluated by measuring the scattering parameters depending on the distance between the antennas. Figure 5 shows S_21_ (black squares), S_11_ (red dots) and S_22_ (green triangles) versus the variation of the distance. The measurements were performed at peak frequencies (since the antennas are resonant), i.e., the shallowest points for S_22_ and S_11_, and the highest ones for S_21_ on the spectra. All frequencies were close to 4.8 GHz. The obtained weak linear dependence of S_21_ on the distance (about −0.18 dB/mm) reflects the possibility of measurements without the strong control of the distance between sensors.

### 2.3. Details of Rhizobox Used

The rhizoboxes were made out of Polymethyl Methacrylate (PMMA, PLEXIGLAS^®^; ε_r_ ≅ 2.6) and Polyvinyl Chloride (PVC; ε_r_ ≅ 2.9) with walls of about 5.0 mm in thickness, a length of 315.0 mm and a width of 200.0 mm. The internal space for the porous media samples was 20.0 mm.

### 2.4. Measurement of Reproducibility and Repeatability

The two chemical solutions used were 1,4-Dioxane (C_4_H_8_O_2_) and 2-Isopropoxyethanol ((CH_3_)_2_CHOCH_2_CH_2_OH), abbreviated as i-C_3_E_1_ [15]. The objective was to make an “ideal solution” by combining water and other liquids fully miscible in water, which will have comparable dielectric properties to soil. Therefore, we prepared mixtures of which concentrations were derived from [15] (see Table 1). The real and imaginary part of the complex dielectric permittivity of the reference liquids can be described by the Cole-Cole model [3]:(1)ε*=ε∞+(εS−ε∞)1+(iffrel)1−α+iσ2πfε0,
where $\varepsilon^{*}$ represents the complex dielectric permittivity, $i=\sqrt{-1}$, $\varepsilon_{S}$ and $\varepsilon_{\infty }$ are the permittivity at frequencies much lower and much higher than the relaxation frequency $f_{rel}$. In turn, α is the dispersion factor for the relaxation time (0 < α < 1), σ is the electrical conductivity, and $\varepsilon_{0}$ is the free space permittivity (8.84 × 10^−12^ F m^−1^). We took values of these parameters from [15] and calculated the real and imaginary part of the complex dielectric permittivity for each obtained mixture. The values shown in Table 1 are used below to evaluate the reproducibility and repeatability of the system (see Section 3.2).

### 2.5. Calibration Curves

For each type of soil, every sample was prepared to have the same bulk density ρ_ss_ of dry soil and its different volumetric moisture, θ_V_. Four different porous media (PM) were used: Nullerde (Einheitserde Typ 0, Einheitserde- und Humuswerke Gebr. Patzer GmbH and Co. KG), peat–sand–pumice substrate (Dachstaudensubstrat SoMi 513, Hawita GmbH, Vechta, Germany Kaktus soil), a dystrophic Red Latosol/Oxisol (Tropical Cerrado soil—Brazil) and glass beads (particles size ～ 0.25 mm). Table 2 shows the physical-chemistry characteristics of the PM samples to conduct the investigation. The $\rho_{ss}$ during the experiments were about 0.28 g/cm^3^, 0.59 g/cm^3^, 1.19 g/cm^3^ and 1.54 g/cm^3^, respectively. The experiments were made under laboratory conditions at with atmospheric temperature of about 25.0 ± 0.5 °C and relative humidity of ~30.0% (see Section 3.4). Additionally, the influence of temperature for Nullerde soil (considered nutrient-deficient substrates) was measured under a climate-controlled environment at temperature range from 13.0 ± 0.5 to 39.0 ± 0.5 °C (see Section 3.3).

The volumetric moisture was taken because the dielectric constant of the soil–water mixture is a function of the water volume fraction in the mixture [16].

The following equation can be used to calculate the volumetric soil moisture, *θ_V_*:(2)$$\theta_{V}\left(\%\right)=\left(\frac{V_{H_{2}O}}{M_{ds}}\times \rho_{ss}\right)\times 100\%$$
where $V_{H_{2}O}$ is the volume of the water (in cm^3^); *M_ds_* is the weight of the dry soil (in g) and $\rho_{ss}$ is the dry soil density (in g/cm^3^).

## 3. Results and Discussion

### 3.1. Measurement of S_21_ under Different Experimental Conditions

The purpose of this experiment was to observe the variation of the attenuation of the S_21_ parameter under the influence of lab air condition, using an empty rhizobox and a rhizobox filled with one porous media (PM; Nullerde organic substrate with one density from dry soil) as well as with DI water. Figure 6 shows the bar graph of S_21_ for the samples that were used. The measurements were taken three times for each sample. The attenuation in the case of DI water was very strong giving S_21_ to be about 26.5 dB, the other samples had S_21_-values close to 9.7 dB. The measured data show a correlation between S_21_ and the dielectric constant of each material, as ε_r_ was changed from 1 (air), to ε_r_ < 5.0 for PMMA/PVC and PM and to 78 for water.

### 3.2. Evaluation of Reproducibility and Repeatability of the System Developed

This experiment aimed to correlate the dielectric permittivity of certain concentrations of organic chemical solutions (see Table 1, in this case mainly 2-Isopropoxyalcohol) with the peak values of S_21_. The results clearly show a correlation between the two variables reflecting that when the dielectric permittivity increases, the attenuation also increases. Figure 7 exhibits the experiment emulating the value of the dielectric constant in water, in terms of repeatability regarding the equipment used. The values presented the average of S_21_ ($\bar{S_{21}}$) = S_21(i)_ ± STD S_21_ for both measured (reproducibility and repeatability) parameters demonstrate the degree of variation with which the obtained results can be regarded, concerning the experimental system that has been developed. (Appendix A are showing the dependences of S21 on different parameters, as (A) tanδ X S_21_ (dB); (B) |ε*| X S_21_ (dB); (C) ε′ X S_21_ (dB) and (D) ε″ X S_21_ (dB)).

### 3.3. Influence of Temperature During the Measurements of Volumetric Soil Moisture

Temperature is an important parameter during the measurements of soil moisture. The following experiment was conducted utilizing a climate-controlled chamber, with temperatures ranging from 13 to 39 °C with 13 °C increments. Figure 8 shows the measured data of S_21_ parameter versus volumetric soil moisture, θ_V_, of the Nullerde samples (soil bulk density ρ_ss_ = 0.28 g/cm^3^) at different temperatures. In dry soil conditions, the signal transmittance shows a slight decrease as the soil temperature increased, which can be explained by the increase of the dielectric loss factor, ε”, for PMMA and PVC. The situation is changed when the PM has a high amount of water. At a temperature difference of about 26 °C, the maximum relative change of S_21_ is about 35% at θ_V_ = 53%. This correlates with the absorbance of the electromagnetic waves by water at 4.8 GHz. As outlined by literature, the complex dielectric permittivity of water, $\varepsilon_{water}^{*}=\varepsilon \prime +i\varepsilon \prime \prime$, changes with temperature [16], having at 13 °C $\varepsilon_{water}^{*}\;≅75+i24.4$, and at 39 °C $\varepsilon_{water}^{*}≅71+i12.5$, which, in terms of dielectric loss tangent, tanδ=ε″/ε′, is 0.325 and 0.176, respectively [17]. Taking into account that θ_V_ is 53%, the rough estimation of the relative change of tanδ gives in this case is about 45%, which is close to the previously mentioned 35%. To calculate more precise values, a mixture model for the soil with water should be considered, taking into account the dielectric permittivity of soil particles [18].

The obtained data indicates a decrease in the efficiency of microwave absorption when the high volumetric moisture soil temperature (> 14.5%) increases, which is mainly driven by the changes in the dielectric properties of water.

The error that can be made when measuring soil moisture at different temperatures cannot be neglected. The indicated temperature effect mainly reflects the changes in the complex soil dielectric permittivity. A similar effect was demonstrated by Gong et al. in 2003 [19] using time domain reflectometry (TDR) sensors. They showed the changes in measurement errors with temperature at different soil moisture levels.

### 3.4. The Calibration Curve and Modeling of Volumetric Soil Moisture θ_V_ as a Function of S_21_

Different calibration curves are needed for different PM. Calibration curves were obtained by measuring the S_21_ parameter, which reflects transmission of EM waves. Measuring the attenuation of the transmitted signal, we obtained clear dependence of θ_V_ (in %) on S_21_ (in dB). The dependences tended to be comparable for different relatively dry PMs, when θ_V_ was below 20%, having small shifts due to different bulk densities, as well as the physical and chemical properties of each PM, especially for the typical tropical soil (Cerrado soil).

The curves obtained for four samples (Cerrado Soil, Nullerde, Kaktus Soil and Glass Beads) are shown in Figure 9 and the following equations presented in Table 3:

In the previous experiment, as shown in Figure 8, it was observed for the Nullerde soil that the influence of the temperature was minimal for volumetric soil moisture around 14.5%. In this way, this value was kept for the soil moisture to evaluate the influence of different PM and its bulk density (**ρ_ss_**) in the measurement of S parameters. Figure 9 shows the effect of the distinct PM in the calibration curve. For the experiment, two distinct types of porous media were used, which exhibited different bulk densities: organic soil substrate (sand substrate at 0.28 g/cm^3^; clay substrate at 1.19 g/cm^3^; siliceous granule substrate at 0.59 g/cm^3^) and inorganic soil substrate (glass beads at 1.54 g/cm^3^). Due to the physical and chemical characteristics of the sample, it was measured that the S21 attenuation was approximately −4.0 dB. These properties must therefore be taken into account for subsequent measurements of volumetric soil moisture (θV) percentage. With the use of time domain reflectometry (TDR) equipment, at frequencies around 1.2 GHz, Gong and collaborators [19] present the effect of soil density.

From Figure 9 it is possible to observe that more water into the porous media (PM) results in a lower S_21_ (the attenuation of the signal increase). As seen in Figure 6, soil did not absorb much energy, and a mixed model could explain this behavior.

The Kaktus soil (SoMi 513) is considered to be a conductive soil, due to it containing various minerals that can be dissolved; therefore, at a certain point of relative moisture, the soil becomes conductive.

With regards to Cerrado Soil (Figure 9), a type of tropical soil, its difference in curve behavior can be attributed to its higher iron content, as well as nitrogen or nitrogen/carbon and their interaction with electromagnetic waves. Von Hippel (1954) [20] shows that permeability of magnetite crystals as a function of frequency (Hz) becomes notably independent when high frequency values are reached. Schmugge (1983) [21] observed in his work that the surface area, in a soil, depends on its particle-size distribution or texture. One of the main features of the clay particle is its large surface area. This allows it to hold more of this tightly bound water than sandy soils; thus, this transition point occurs at higher moisture levels in clay than in sandy soils.

Figure 10 shows preliminary results of the calculation of $\bar{S_{21}}$ attenuation of volumetric soil moisture in the rhizobox during the time of 16 days and 21 h and 17 min. It is possible to observe how the values of S_21(i)_ and STD S_21_ change with volumetric soil moisture, during the elapsed time inside the rhizobox. The sample is Kaktus soil with a maize plant. The experiment was organized to observe the attenuation of soil moisture during the time of root growth. The measurements were taken in laboratory conditions (Temperature (T(°C)) = 24.3 ± 1.1 °C and Relative Humidity (RH(%)) = 62.8% ± 5.0%) during the morning, the afternoon, at night and at dawn, to measure the behavior of the water. To conduct this experiment three distinct steps were done: (1) start condition using just soil moisture inside the rhizobox; (2) included two maize seeds into soil and (3) after 12 days and 9 h and 11 min 30 mL of DI water was added.

Initially, the experiment was fed with water at Steps 1 and 2. Subsequently, the plant was left to grow and the soil was left to dry out for 150 h, with measurements taken at 6 h intervals (morning—9:00 am; afternoon—3:00 pm; night—9:00 pm and dawn—3:00 am). The quantitative data was obtained from a specific area of 168 cm^2^, around 48.0% of the total area of the rhizobox with soil. To calculate the $\bar{S_{21}}\;=S_{21\left(i\right)}$ ± STD S_21_, i = 1173 data points were used.

The volumetric soil moisture θ_V_ decreased with time, from 34.5% to 27.2%. The range of the data were collected after 1 day, 15 h and 17 min until 12 days, 9 h and 11 min from the experiment running, and were showing the increase of attenuation measuring the S_21(i),_ calculated the $\bar{S_{21}}$ and STD S_21_. Table 4 presents a specific region from Figure 8 that shows an inversion behavior of average $\bar{S_{21}}$ (dB) and used the absolute value of the STD S_21_ (dB).

As the dielectric constant of the root (42 < εr < 56 (f_0_ = 1.2 GHz)) was much higher than that of dry soil (3 < εr < 8) and nearer to that of water (εr ~79) [7], it was possible to hypothesize that it could be a factor of root growth influence, mainly with an increase of the root density in the regions where the measurements were taken. This approach was undertaken due to certain established experimental conditions and presumptions: due to the soil substrate density of this experiment being relatively low (ρss = 0.59 g/cm^3^), we postulated that there were more porous gaps in the medium, which therefore would influence the permeability of water and the roots, in accordance to the behavioral model of the triphasic mixture [22,23] in regards to the environment, which was taken into account in this study. De Willigen et al. [24] established that the hydraulic redistribution of water caused by root dynamics maintains root turgidity. Bao et al. [6] demonstrates the importance of the presence of water around the tip of the root, as this is shown to play an important role in the formation of the root system. In this manner, we inferred that for the area of interest, the root and the fraction of water likely occupy the larger porous proportions in the soil substrate and consequently influence the dielectric constant in calculations involving soil media and S_21_ standard deviation.

The experiment shown in Figure 10 is considered a new opportunity to study non-invasive phenotyping and the influence of soil and water in root and plant growth against a measure of time using a non-radioactive electromagnetic wave, as well as an insertion rod that did not disturb the root–soil matrix.

## 4. Conclusions

In this work from the measured results it is possible to conclude that the developed non-invasive microwave method, using microstrip antennas, is an innovative sensing method to measure the water status in a rhizobox filled with soil. The system was developed without the utilization of invasive rods—similar to the TDR technique, which is normally used in the measurement of soil moisture—which could have an influence in the direction of root growth. Low power non-ionizing microwave radiation was used, with the frequency around 4.8 GHz using attenuation measurements. The λ/2- resonant feedback microstrip antennas used in this endeavor presented the proper transmittance and received energy for the measurement of S-parameters. The strong attenuation S_21_ (dB) of the DI water was significant when compared with different materials and dry soil, thus showing the dependence of the dielectric permittivity of water. Part of the water–PM mixture is dominated by the higher permittivity of the water at the microwave frequency that was used. The results presented demonstrate that the system developed has a high reproducibility rate (98.9%) and a satisfactory repeatability rate (93.0%) to measure volumetric soil moisture θ_V_. The influence of temperature on the measurement of θ_V_ was discussed. Since the temperature affects the error of the measurements of volumetric soil moisture, a calibration curve requires the information of both soil temperature and soil water content. The distinct effect of the PMs on the calibration curve (S_21_ vs θ_V_) was also observed, giving an opportunity to use such an approach to investigate the growth of plant roots together with soil physical properties.

## Figures and Tables

**Figure 1 sensors-20-01166-f001:**
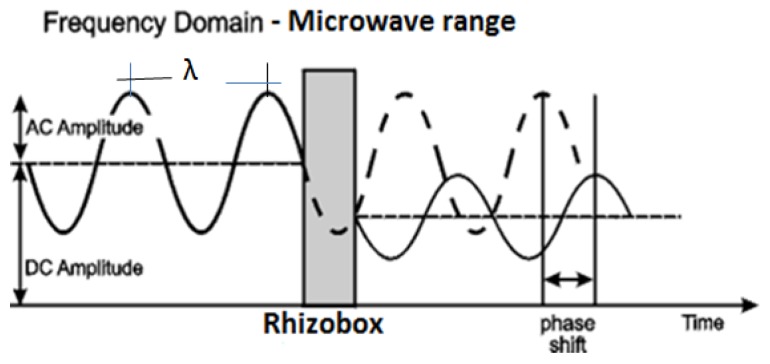
Diagram with the basic principle of the electromagnetic wave interactions with a rhizobox filled by a porous medium (attenuation and phase shift; modified from [11]).

**Figure 2 sensors-20-01166-f002:**
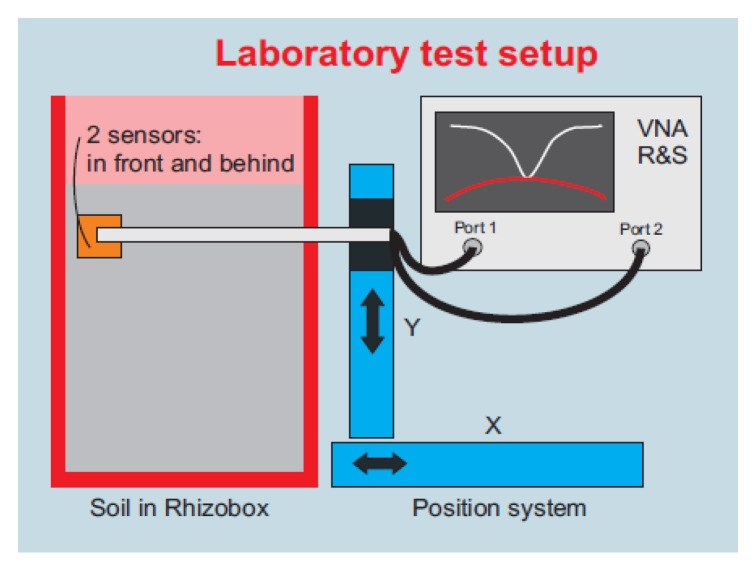
The block diagram of the system developed to measure S_21_ (dB) of the soil moisture in the rhizobox, using the Vector Network Analyzer, in the microwave frequency range (4.6–5.0 GHz).

**Figure 3 sensors-20-01166-f003:**
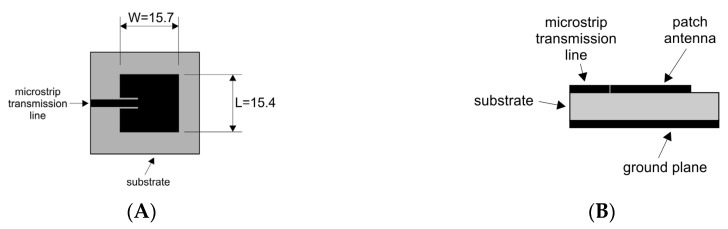
The mask used to develop the microstrip patch antenna on the circuit board. (**A**) Top view of the patch antenna and (**B**) side view of the microstrip antenna.

**Figure 4 sensors-20-01166-f004:**
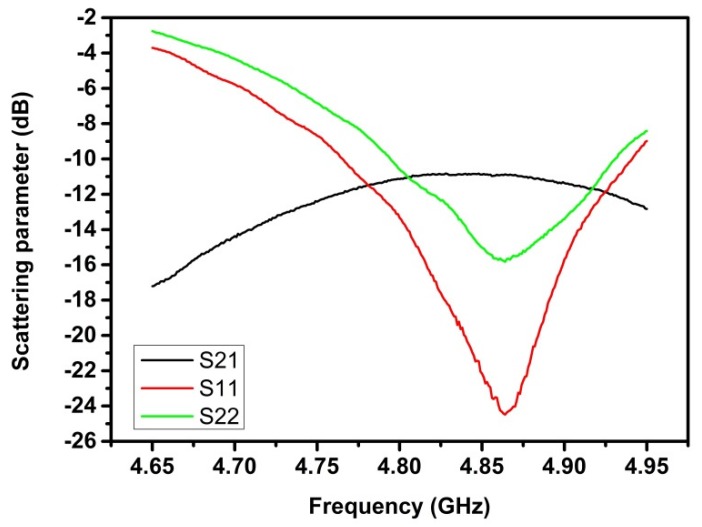
Magnitudes of the scattering parameters S_21_ (black line), S_11_ (red line) and S_22_ (green line) versus the frequency for the microstrip antenna (MSA) used in this work when the distance between antennas was 24 mm. The measurements were obtained under standard laboratory atmospheric conditions.

**Figure 5 sensors-20-01166-f005:**
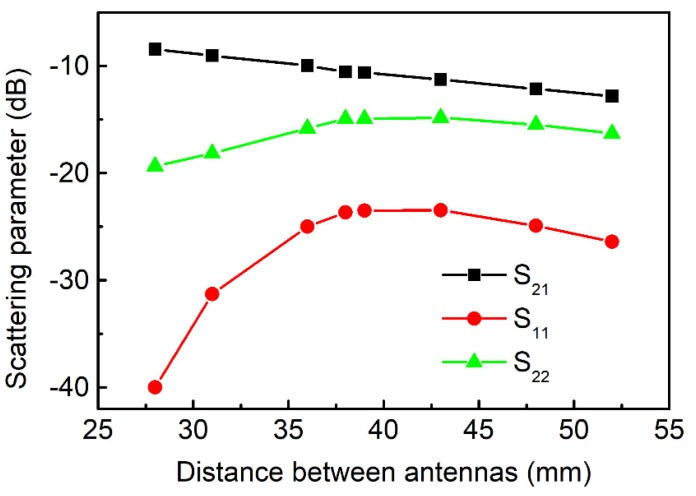
Scattering parameters S_21_ (black squares), S_11_ (red dots) and S_22_ (green triangles) versus the distance between the antennas. Measurements are done at peak frequencies close to 4.8 GHz, i.e., the shallowest points for S_22_ and S_11_, and the highest ones for S_21_.

**Figure 6 sensors-20-01166-f006:**
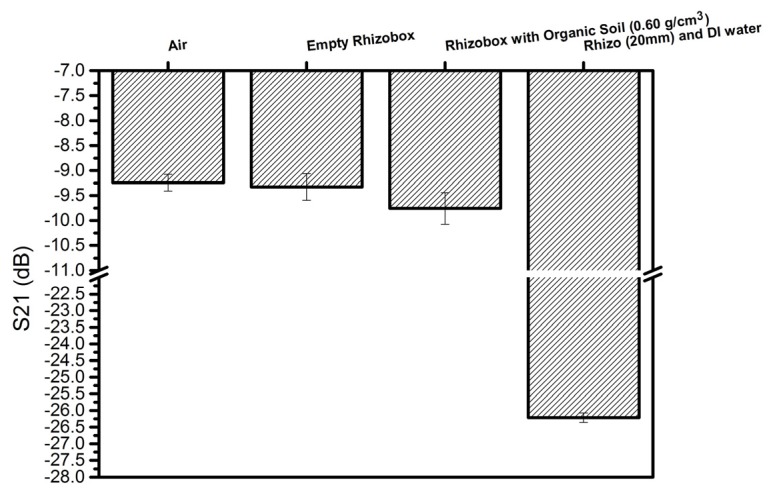
The peak values of scattering parameter, S_21_, measured for the following samples: lab air (Air), empty rhizobox, the rhizobox filled with one of the porous media (substrate Nullerde, an organic soil with density $\rho_{ss}$ = 0.60 g/cm^3^) and DI water. The frequency used in this experiment was close to 4.8 GHz.

**Figure 7 sensors-20-01166-f007:**
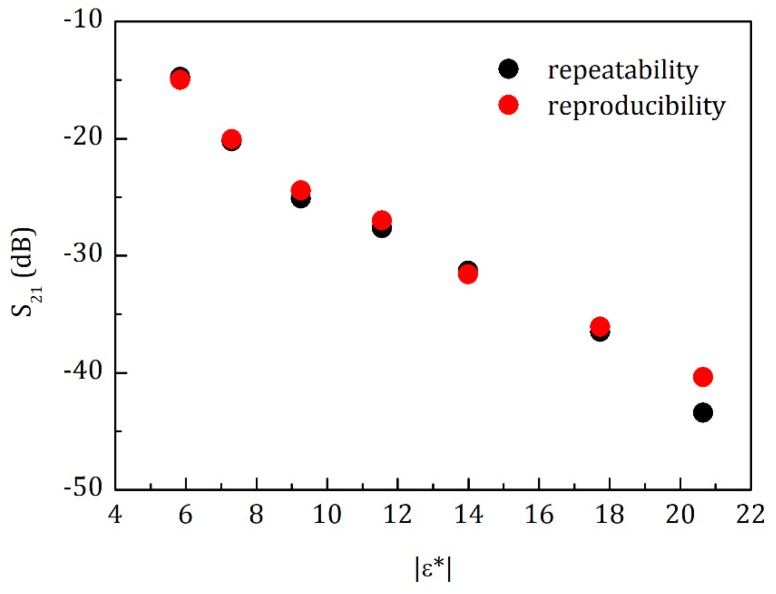
The repeatability and reproducibility of the system developed were calculated. The measurement was carried out three times to each dot (*n* = 3) using concentrations of organic chemical solutions from Table 1 and the relation between $\left|\varepsilon^{*}\right|$ versus the average of $S_{21}\;\left(dB\right)$. The red dots represent the reproducibility (98.9%) averages and the black dots represent the repeatability (93.0%) averages.

**Figure 8 sensors-20-01166-f008:**
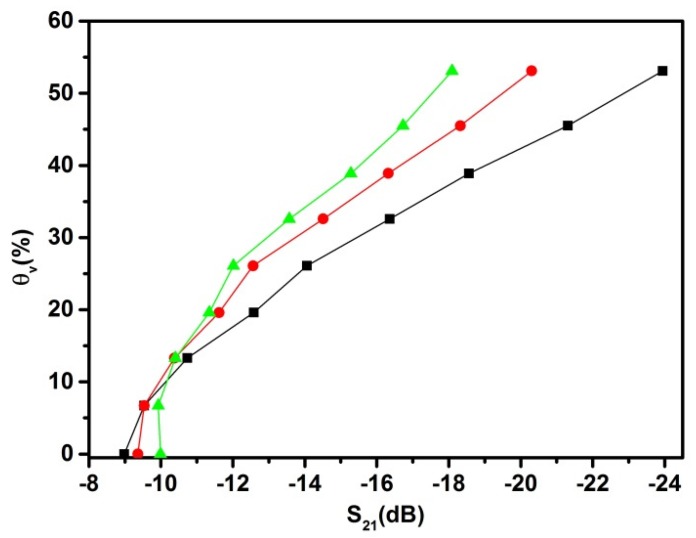
Influence of temperature on the θ_V_ measurements using attenuation S_21_. The porous medium used was Nullerde, and the temperature set up for this experiment was 13.0 ± 0.5 (black squares), 26 ± 0.5 °C (red dots) and 39.0 ± 0.5 °C (green triangles).

**Figure 9 sensors-20-01166-f009:**
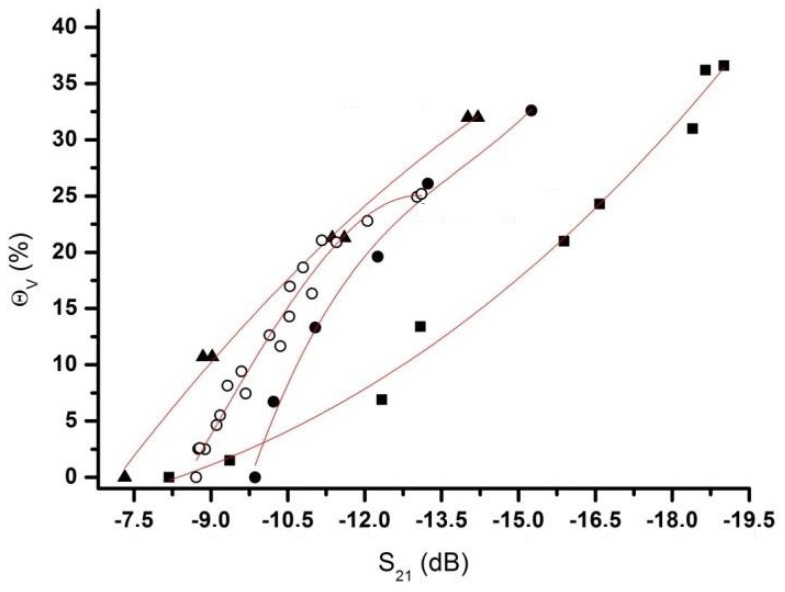
Relationship between the S_21_ measured with the developed system and the volumetric soil moisture θ_V_ determined and calculated by the second and third order polynomial equation. The equations are presented in Table 3. The four (04) samples used are Cerrado Soil (squares), Nullerde (dots), Kaktus Soil (open circles) and Glass Beads (triangles). The experiment was carried out at standard laboratory ambient conditions (Temperature (T(°C)) = 25.0 ± 0.5 °C and Relative Humidity (RH(%)) ≅ 30%).

**Figure 10 sensors-20-01166-f010:**
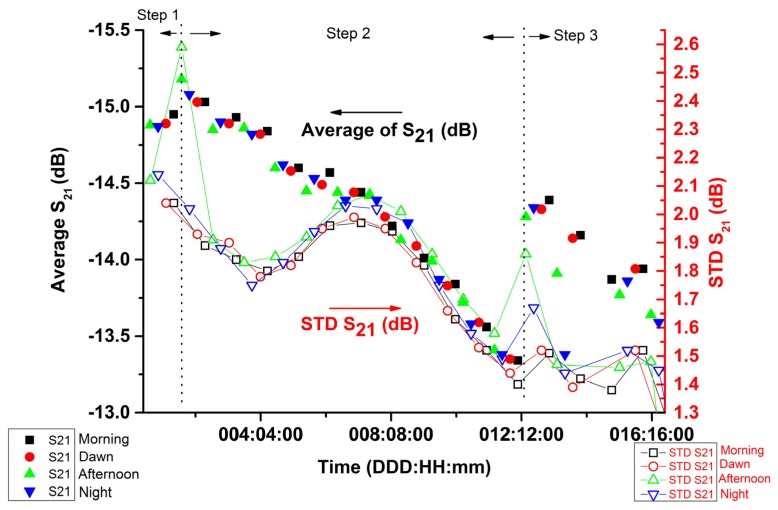
Results of the measurement of $\bar{S_{21}}$ attenuation that correlate with volumetric soil moisture in the rhizobox, during the time of 16 days, 21 h and 17 min and a calculation of standard deviation (STD S_21_). The sample is the Kaktus soil with maize. The experiment was organized in three steps to measure the attenuation of soil moisture during the time of root growth. The measurement was taken in standard laboratory conditions (Temperature (T(°C)) = 24.3 ± 1.1 °C and Relative Humidity (RH(%)) = 62.8% ± 5.0%) at dawn (3:00 am), in the morning (9:00 am), in the afternoon (3:00 pm) and at night (9:00 pm).

**Table 1 sensors-20-01166-t001:** Calculated dielectric properties of the test liquids at 25 °C and f = 4.8 GHz using (1) and data shown in [16].

Medium	Volume Fraction in Deionized Water (DI Water)	$\varepsilon^{\prime }$	$\varepsilon^{\prime \prime }$	tanδ	$\left|\varepsilon^{*}\right|=\sqrt{\varepsilon^{\prime 2}+\varepsilon^{\prime \prime 2}}$
	**(cm^3^/cm^3^)**	(-)	(-)	(-)	(-)
Dioxane	0.90	5.76	0.97	0.168	5.84
i-C_3_E_1_	1.00	6.71	2.87	0.428	7.30
i-C_3_E_1_	0.92	7.91	4.82	0.609	9.26
i-C_3_E_1_	0.86	9.51	6.57	0.691	11.56
i-C_3_E_1_	0.80	11.22	8.36	0.745	13.99
i-C_3_E_1_	0.73	14.03	10.84	0.773	17.73
i-C_3_E_1_	0.68	16.42	12.53	0.763	20.66

**Table 2 sensors-20-01166-t002:** The physical–chemistry characteristics of the porous media (PM) samples used.

Samples	Clay	Sand	Silt	N	C	C/N	Fe	$\rho_{ss}$
	%	%	%	%	%		mg/g	g/cm^3^
Nullerde (Einheitserde Typ 0)	35.0	-	-	0.45	22.1	48.9	30.5	0.28
Dachstau den substrat Somi 513 (Kaktus soil)	19.1	-	23.1	0.21	12.5	43.0	183	0.59
Dystrophic Red Latosol (Oxisol)—Cerrado Soil (Go/BR)	58.2	18.4	23.4	-	18.5	-	120	1.19
Glass Beads * (210–250 µm)								1.54

* Glass Beads chemical composition (by weight): Silica (SiO_2_) = 66–75%; Aluminum Oxide (Al_2_O_3_) = 0–5%; Calcium Oxide (CaO) = 6–15%; Magnesium Oxide (MgO) = 1–5%; Sodium Oxide (Na_2_O) = 10–20% and Iron Oxide (Fe_2_O_3_) = < 0.8%.

**Table 3 sensors-20-01166-t003:** Table with four equations used to modeling the calibration to each sample.

n^0^	Samples	Equation	* R^2^
(3)	Cerrado Soil ρ_ss_ = 1.19 g/cm^3^	Θ_v_(%) = −0.00258 * S21(dB)3 + 0.07924 * S21(dB)2 + 0.27515 * S21(dB) − 4.72555	0.9834
(4)	Null Erde ρ_ss_ = 0.28 g/cm^3^	Θ_v_(%) = −0.18679 * S21(dB)3 − 7.79899 * S21(dB)2 − 112.0218 * S21(dB) − 524.374	0.9804
(5)	Kaktus Soil ρ_ss_ = 0.59 g/cm^3^	Θ_v_(%) = 0.1555 * S21(dB)3 + 4.16601 * × S21(dB)2 + 29.24777 * S21(dB) + 42.96299	0.9725
(6)	Glass Beads ρ_ss_ = 1.54 g/cm^3^	Θ_v_(%) = −0.19723 * S21(dB)2 − 8.78181 * S21(dB) − 52.87515	0.9923

* The Coefficient of Determination (R^2^) is the proportion of the variance in the dependent variable that is predictable from the independent variable(s).

**Table 4 sensors-20-01166-t004:** Values obtained from specific region of Figure 8.

	Time (DD:HH:mm)	$\bar{S_{21}}$ (dB)/θ_V_ (%) ^1^	STD S_21_ (dB)
**Start**	03:21:10	−15.18/34.5	1.75
**End**	06:20:50	−14.39/27.2	2.03

^1^ θ_V_ was calculated using the Equation (5) Kaktus soil, from Table 3.

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
