# Peer review of "Microwave Transmittance Technique Using Microstrip Patch Antennas, as a Non-Invasive Tool to Determine Soil Moisture in Rhizoboxes†"

_sensors, 2020, doi:10.3390/s20041166_

Round 1

Reviewer 1 Report

See attached file

Author Response

To Dr. Stanley Bao

Managing Editor

Sensors – MDPI

Ms. Ref. No.: sensors-679195

Title: Microwave transmittance technique using microstrip patch antennas, as a non-invasive tool to determine soil moisture in rhizoboxes

Dear Dr. Bao,

Please find uploaded a revised version of the manuscript entitled "Microwave transmittance technique using microstrip patch antennas, as a non-invasive tool to determine soil moisture in rhizoboxes.". The authors have revised and modified the manuscript according to the reviewers’ comments. A detailed answer for each reviewer comments is attached to this letter (Please see the attachment). The authors would like to thank the reviewers for their valuable comments, which helped improve the manuscript's quality. The changes made are highlighted in the manuscript. We believe that all doubts have been clarified and that the revised manuscript should be now suitable for publication in Sensors/MDPI.

I appreciate your time and consideration.

Your sincerely,

Dr. Paulo Sergio de Paula Herrmann Junior

Senior Researcher

Embrapa Instrumentation

Rua XV de novembro 1452

Zip Code: 13560-741

São Carlos, SP - BRAZIL

 +55 16 2107 2911(office)

 +55 16 2107 2805 (Secret)

Reviewer 2 Report

In this paper, the authors showed the possibility of determination of soil moisture in rhizoboxes based on a microwave transmittance technique using a microstrip patch antenna (MPA).From the viewpoint of the experiment, this paper was well written and systematically organized.

However, I still have some questions in this work:

The system for determination of soil moisture is based on microstrip patch antenna (MPA). Is MPA most properly antenna? The authors conducted the experiment at specific frequency of 4.8 GHz. Is this frequency also reliable in the system? Is this experimental system the same environment of Ref. [13]? As the authors mention the manuscript, the soil-plant-water set consists of air, soil volume, bound water, free water and roots. This system is theoretically analyzed by Cole-Cole model. Actually, the soil-plant-water consists of five different dielectric constants. Is this model validated for mixture model?

In addition, the image resolution of figures should be improved in the manuscript.

Author Response

(The authors gave the same response as above.)

Reviewer 3 Report

The article shows an interesting application of microwaves. The manuscript is well written and is clear.

I have only some minor suggestions:

- line 254: Actually when the water increases the S21 parameter decreases.

- Why in the table 3 the first column contains numbers from 3 to 6?

- Please define the quantity R^2 in table 3

Author Response

(The authors gave the same response as above.)

Round 2

Reviewer 2 Report

The authors have well reflected my comments and curiosities faithfully in the revised version.

So I recommend that this manuscript is acceptable for publication in Sensors Journal.